# High Resolution Powder Electron Diffraction in Scanning Electron Microscopy

**DOI:** 10.3390/ma14247550

**Published:** 2021-12-09

**Authors:** Miroslav Slouf, Radim Skoupy, Ewa Pavlova, Vladislav Krzyzanek

**Affiliations:** 1Institute of Macromolecular Chemistry of the Czech Academy of Sciences, Heyrovsky Sq. 2, 16206 Prague, Czech Republic; pavlova@imc.cas.cz; 2Institute of Scientific Instruments of the Czech Academy of Sciences, Kralovopolska 147, 61264 Brno, Czech Republic; radim.skoupy@isibrno.cz

**Keywords:** nanoparticle analysis, powder nanobeam electron diffraction, 4D-STEM

## Abstract

A modern scanning electron microscope equipped with a pixelated detector of transmitted electrons can record a four-dimensional (4D) dataset containing a two-dimensional (2D) array of 2D nanobeam electron diffraction patterns; this is known as a four-dimensional scanning transmission electron microscopy (4D-STEM). In this work, we introduce a new version of our method called 4D-STEM/PNBD (powder nanobeam diffraction), which yields high-resolution powder diffractograms, whose quality is fully comparable to standard TEM/SAED (selected-area electron diffraction) patterns. Our method converts a complex 4D-STEM dataset measured on a nanocrystalline material to a single 2D powder electron diffractogram, which is easy to process with standard software. The original version of 4D-STEM/PNBD method, which suffered from low resolution, was improved in three important areas: (i) an optimized data collection protocol enables the experimental determination of the point spread function (PSF) of the primary electron beam, (ii) an improved data processing combines an entropy-based filtering of the whole dataset with a PSF-deconvolution of the individual 2D diffractograms and (iii) completely re-written software automates all calculations and requires just a minimal user input. The new method was applied to Au, TbF_3_ and TiO_2_ nanocrystals and the resolution of the 4D-STEM/PNBD diffractograms was even slightly better than that of TEM/SAED.

## 1. Introduction

Scanning electron microscopy (SEM) is a well-established method for characterization of materials in both micro- and nanoscale. The classical modes of SEM comprise imaging with secondary electrons (SE), backscattered electrons (BSE) and transmitted electrons (scanning transmission electron microscopy; STEM), and microanalysis (energy-dispersive analysis of X-rays; EDX). In the field of STEM, the standard detectors collect just the integral signal from the electrons going through the specimen almost directly (bright field imaging, BF) or from the electrons scattered at higher angles (annular dark field imaging, ADF, and high-angle annular dark field imaging, HAADF).

The standard STEM/BF, STEM/ADF and STEM/HAADF micrographs are two-dimensional (**2D-STEM**), i.e., every XY-position on the sample gives one signal on the detector [1]. However, the fact that STEM imaging might yield more information, on the condition that it was possible to record positions of the scattered electrons, did not escape the attention of researchers. An intermediate step was the commercialization of multi-segmental STEM detectors, which consisted of one central segment in BF region and multiple annular (or semi-annular) segments in ADF and HAADF regions [2,3]. For each XY-position on the sample, the multi-segmental STEM detector can yield signal from each of its segments. This is occasionally called **multidimensional STEM**, where the additional dimension is the angular resolution [4]. The final step was the recent commercialization of 2D-array detectors (also known as pixelated detectors). The pixelated STEM detectors record both the intensity and position of the scattered electrons, i.e., they record a 2D diffraction pattern for each beam position. If the scanning beam is narrow and non-convergent, the recorded pattern corresponds to nanobeam diffraction (NBD) known from TEM (but the classical TEM/NBD measurements are performed with a fixed, non-scanning beam [5]). As the diffraction patterns are recorded for all XY-positions of the primary beam on the specimen, we get a 4D data cube (a 2D array of 2D diffraction patterns) and the technique is referred to as **4D-STEM**. The traditional pixelated STEM detectors are indirect, combining a scintillator (a material converting ionizing radiation to light) with a digital camera based on CCD (charge coupled device) or CMOS (complementary metal oxide semiconductor) technology; these devices suffer from somewhat lower speed and/or higher noise. The modern pixelated STEM detectors are direct (DED; direct electron detectors) and exhibit higher speed, lower noise, and smaller size in comparison with traditional CCD and CMOS cameras [6,7].

The 4D-STEM techniques were introduced in the field of TEM and dedicated high-energy STEM microscopes, where they have already found numerous applications as summarized in recent reviews [8,9]. In the field of SEM, the 4D-STEM techniques are much less common. The fast pixelated DED detectors for SEM were commercialized quite recently [10]. Consequently, just a few examples of 4D-STEM in SEM can be found in the literature, and most of the studies were based on traditional CCD and CMOS detectors [11,12,13,14]. This resulted in somewhat non-standard hardware solutions, in which bulky CCD and CMOS cameras had to be attached to adjusted SEM columns. Moreover, the authors focused their attention on the analysis of the individual diffraction patterns of the 4D dataset, which required detailed crystallographic knowledge and special software. In contrast, our recent study [15] was focused on simple, fast, and efficient usage of 4D-STEM method in everyday life. In our method, which was called 4D-STEM/PNBD (powder nanobeam diffraction), we combine all diffraction patterns (showing diffraction spots from one or more nanocrystals) into a single powder diffraction pattern (showing diffraction rings typical of polycrystalline samples). In this way, we converted the complex 4D dataset into a simple 2D powder diffractogram. We analyzed several sets of inorganic nanocrystals deposited on thin carbon film and the 4D-STEM/PNBD diffraction patterns corresponded to the well-known TEM/SAED (selected-area electron diffraction) patterns. Consequently, the 4D-STEM/PNBD and TEM/SAED diffractograms could be processed with the same, well-established programs, such as *ProcessDiffraction* [16]. These programs do not require deep crystallographic knowledge, as the processing of powder diffractograms is quite straightforward. The measurements were performed with a high-resolution SEM microscope equipped with a fast DED detector. The small DED detector can be mounted in any modern SEM microscope with a standard port for STEM, which makes the method accessible to all users of modern SEM’s.

In this contribution, we have resolved the most important limitation of the previous version of our 4D-STEM/PNBD method, which consisted of the lower-intensity, broader diffraction peaks in comparison with TEM/SAED. The current version 4D-STEM/PNBD method has been improved in three important aspects: At first, we optimized the measurement protocol so that we could get the accurate point spread function (PSF) of the primary beam. At second, we combined entropy-based filtering of the whole 4D dataset with PSF-deconvolution of the individual 2D diffractograms, which lead to a significant enhancement of both intensities and resolution of the diffraction peaks. At third, we upgraded our Python package STEMDIFF, which automates all 4D-STEM/PNBD calculations. The current version of the program includes both entropy-based filtering and PSF deconvolution, various minor improvements and better documentation, and easier usage that requires just very minimal user input. Moreover, the original STEMDIFF scripts were rewritten and converted into a standardized distributable package, which can be installed from the official internet Python Package Index (PyPI; *https://pypi.org*; accessed on 15 November 2021). The improved 4D-STEM/PNBD method was applied to Au, TbF_3_ and TiO_2_ nanocrystals with two different crystalline modifications (anatase and rutile). In all four cases, the crystals could be unambiguously identified and the resolution of the 4D-STEM/PNBD diffractograms was the same or even slightly better in comparison with the corresponding TEM/SAED patterns. We summarize that the current version of 4D-STEM/PNBD method can analyze polycrystalline samples in SEM microscopes in a fast and easy way, and the quality of the results is fully comparable with the standard TEM/SAED method.

## 2. Materials and Methods

### 2.1. Samples

Four nanocrystalline samples with various average crystal sizes were selected for testing of the improved 4D-STEM/PNBD method. In all four cases, the nanocrystals were deposited on a standard TEM copper grid coated with an electron-transparent carbon film. The first samples were Au nanoislands (size ~ 20 nm); their preparation was described in our previous study [17], in which they were employed as a nucleating agent for polypropylene. The second samples were small TbF_3_ nanocrystals (size < 5 nm) with admixture of Gd^3+^, Yb^3+^ and Nd^3+^ ions) that were designed as multimodal contrast agent for down- and upconversion luminescence, magnetic resonance imaging, and computed tomography [18]. The last two samples were commercial TiO_2_ nanoparticles with two different crystalline modifications: anatase (average crystal size 25 nm) and rutile (average size 80 nm). Both anatase and rutile nanopowders were bought from Sigma-Aldrich (Prague, Czech Republic), dispersed in distilled water, sonicated (20 min) and a small droplet of the sonicated solution was put onto a carbon film and left to dry. The carbon-coated TEM grids with deposited nanoparticles were used for all following experiments.

### 2.2. TEM Characterization

All four samples (Au nanoislands [17], TbF_3_ nanocrystals [18], TiO_2_/anatase, and TiO_2_/rutile) were thoroughly characterized by means of TEM microscopy (microscope Tecnai G2 Spirit Twin; FEI, Brno, Czech Republic; accelerating voltage 120 kV). Briefly, the nanocrystal morphology was visualized by bright field imaging (TEM/BF), elemental composition was assessed from energy-dispersive X-ray analysis (TEM/EDX) and crystalline structure was confirmed using selected-area electron diffraction (TEM/SAED). The final polycrystalline, powder TEM/SAED patterns were processed and converted to radially-averaged 1D-diffraction profiles by means of *ProcessDiffraction* program [16].

### 2.3. Calculation of PXRD Diffraction Patterns

The experimental TEM/SAED diffraction patterns were compared with theoretical powder X-ray diffractograms (PXRD). The PXRD diffractograms were calculated with program *PowderCell* [19]. The calculations were performed for perfect polycrystalline sample with random orientation of crystallites. The crystal structures for the calculation were obtained from the *Crystallography Open Database* [20].

### 2.4. 4D-STEM/PNBD Measurements and Calculations

#### 2.4.1. SEM Microscope with Pixelated Detector

The 4D-STEM/PNBD method yields powder electron diffraction patterns by means of arbitrary SEM microscope equipped with a pixelated STEM detector. We employed a focused ion beam scanning electron microscope Helios G4 HP (FIB-SEM microscope; Thermo Fisher Scientific, Waltham, MA, USA) equipped with pixelated STEM detector T-pix (Thermo Fisher Scientific, Waltham, MA, USA), which was based on Timepix technology (DED detector with the array of 256 × 256 pixels, pixel size 55 μm and intensity detected in range 0–11,810 counts-per-pixel [21]). All measurements were performed at accelerating voltage of 30 kV; more details about the microscope can be found in our previous study [15].

#### 2.4.2. Principle of 4D-STEM/PNBD Method

The principle of 4D-STEM/PNBD method is shown in Figure 1. The nanobeam diffraction patterns (NBD) and powder nanobeam diffraction patterns (PNBD) in Figure 1 come from the real measurement of sample TbF_3_ (the sample is described in Section 2.1).

In a 4D-STEM measurement, the electron beam (primary beam) scans a 2D-array on a specimen and each XY-position of the beam gives a 2D-NBD pattern on pixelated detector (as shown schematically in left part of Figure 1). We note that Figure 1 shows just four beam positions and four corresponding NBD patterns for the sake of simplicity, while a typical scanning array during 4D-STEM measurements contains at least 2000 beam positions. The key step of 4D-STEM/PNBD method is the combination of the individual NBD’s into the final PNBD pattern (as shown in the right part of Figure 1). The straightforward summation of all NBD patterns usually results in low-quality diffractograms, where diffractions are almost lost in the background noise and scattering of the amorphous carbon supporting film (Figure 1a). Better PNBD patterns are obtained if we sum just the NBD patterns containing strong diffractions (Figure 1b). The patterns with strong diffractions are selected from the dataset easily, as they exhibit high values of Shannon entropy [15]. Nevertheless, the best PNBD patterns are achieved if we calculate point spread function (PSF) of the primary beam (Figure 1c) from low-entropy NBD files and then we sum the high-entropy NBD patterns after the application of PSF deconvolution (Figure 1d). The implementation of PSF deconvolution is one of the key improvements of our 4D-STEM/PNBD method, as described and exemplified in this study.

#### 2.4.3. 4D-STEM/PNBD Measurements

The 4D-STEM/PNBD method consists of two basic steps: 4D-STEM measurement (left part of Figure 1) and 4D-STEM/PNBD calculations (right part of Figure 1). The first step comprises three sub-steps: (i) the acquisition of standard STEM/BF micrograph in order to visualize the nanoparticles of interest, (ii) the definition of scanning array (scanning matrix) in the STEM/BF micrograph, from which we will obtain individual NBD patterns, and (iii) the 4D-STEM measurement itself, i.e., the point-by-point scanning of the selected 2D matrix and saving of the individual 2D-NBD diffractograms. The STEM/BF micrograph can be obtained either with a standard STEM detector (if available in a given system) or with the pixelated STEM detector (if we take the integral signal just from the central pixels of the detector, i.e., from the central circle with diameter ~ 30 pixels). The parameters for the 4D-STEM data collection can be defined conveniently with a control software of the pixelated detector. The key data collection parameters are the *scanning step* (=distance between two scanning points), *scanning matrix* (=area from which we will collect NBD diffractograms), and *exposure time* (=time per measurement of one NBD pattern). As for the *scanning step* parameter, we select optimal size according to the typical crystal size observed in STEM/BF micrograph: too small step would lead to multiple data collections from one crystal (redundant data) and too big step would result in too large scanning areas (which would force us to introduce corrections for possible beam shifts; in our systems the beam shifts were found to be insignificant on condition that the scanning area size < 10 μm). As for the *scanning matrix*, we usually use a rectangular array with at least 2000 points, but the shape can be arbitrary. As for the *exposure time*, we use simple trial-and-error method to set the time in such a way that the maximum intensity of the primary beam was slightly below the detector upper limit (i.e., 11,810 counts per pixel in the case of T-pix detector, as specified in Section 2.4.1). The correct exposure time setting is very important. Higher exposure times would not damage the detector, but the overflown intensity of the central spot would prevent us from determining the precise primary beam shape (i.e., primary beam PSF in the XY plane) and, as a result, we could not perform the correct PSF deconvolution. Lower exposure times would result in weak diffractions, whose intensity might be too low for the correct background subtraction. The parameters of data collections used in this study are given in Table 1.

Additional technical details concerning the measurement can be found in our previous work [15]. Once the measurement parameters are set, the final 4D-STEM measurement is performed automatically using the simple control software coming with the pixelated detector. During the measurement, the individual diffraction patterns are saved one-by-one to the selected directory in the form of binary files and the whole data collection time usually does not exceed 12 min.

The second step of 4D-STEM/PNBD method—the summation of the individual NBD patterns in order to obtain the final PNBD pattern—has been automated by means of our freeware program package called STEMDIFF. The details are given below in the Results section, as the updated STEMDIFF package is one of the significant outputs of this work, and, more importantly, the data processing is closely associated with the final results. Here we just briefly summarize that the updated STEMDIFF package requires a minimal user input and performs all types of summations, i.e., the direct summation of all NBD files (Figure 1a), the summation of selected, high-entropy NBD files (Figure 1b), and the summation of high-entropy files with PSF deconvolution (Figure 1c,d). The final 4D-SEM/PNBD diffraction patterns (Figure 1a,b,d) can be processed with arbitrary software for powder diffraction pattern analysis. The most straightforward option is to process the 4D-STEM/PNBD patterns with the same software that is used for the analysis of TEM/SAED patterns, such as the well-established *ProcessDiffraction* program (for the processing of experimental powder electron diffraction patterns [16]) and *PowderCell* program (for the calculation of theoretical powder X-ray diffraction patterns [19]).

## 3. Results

### 3.1. Results of the Improved 4D-STEM/PNBD Method

The improved 4D-STEM/PNBD method was applied on four samples. The first two samples (Au nanoislands and TbF_3_ nanoparticles) had already been studied with the previous version of 4D-STEM/PNBD and we tested how the results could be improved with the current version of the method. The last two samples (TiO_2_ nanoparticles with different crystalline modifications—anatase and rutile) were newly prepared in order to demonstrate that the 4D-STEM/PNBD method could distinguish samples with identical chemical composition.

#### 3.1.1. Au Nanoislands: Strongly Diffracting Nanocrystals

The TEM and 4D-STEM/PNBD results for Au nanoislands are summarized in Figure 2. The Au nanoislands on thin carbon film (Figure 2a) represent quite large nanocrystals, which exhibited very strong diffractions not only in TEM/SAED pattern (Figure 2b), but also in 4D-STEM/PNBD pattern calculated without PSF deconvolution (Figure 2c).

We processed the Au dataset with the recent version of 4D-STEM/PNBD method in order to demonstrate how it can improve the results in the case of strongly diffracting nanocrystals. In the first step, we reproduced the calculation from our previous study (i.e., we used the newly measured dataset and updated software, but we did not employ PSF deconvolution) and the resulting 4D-STEM/PNBD diffractogram was quite comparable to TEM/SAED (cf. Figure 2b,c). Nevertheless, the PSF deconvolution yielded even better diffractogram, with stronger and sharper diffraction rings (Figure 2d).

The improvement of the 4D-STEM/PNBD results due to PSF deconvolution is best documented on 1D-diffraction profiles (Figure 2e). The standard PNBD calculation (i.e., summation without PSF deconvolution; orange line) gave reasonable diffractogram (orange line), corresponding to theoretical PXRD pattern (blue line), but the resolution (i.e., the diffraction widths) was definitely worse in comparison with TEM/SAED (black dotted line). The PNBD calculation with PSF deconvolution (red line) yielded the diffraction pattern, in which the diffraction intensities and resolution were even slightly better than in the case of TEM/SAED (namely in the region of high-angle diffractions with *q* > 5 Å^−1^). This could be attributed to a positive side effect of PSF deconvolution, which decreases the background intensity and improves signal-to-noise ratio (compare Figure 2b–d).

#### 3.1.2. TbF_3_ Nanoparticles: Smaller Nanocrystals with Preferred Orientation

The results for TbF_3_ nanoislands are summarized in Figure 3. The TbF_3_ nanocrystals (Figure 3a) were small and tended to form agglomerates. They exhibited strong diffractions in TEM/SAED (Figure 3b), but their 4D-STEM/PNBD pattern calculated without PSF deconvolution showed quite weak and broad diffraction rings (Figure 3c).

The weaker diffraction power of TbF_3_ nanoparticles in 4D-STEM (Figure 3c) was connected with their smaller size (weaker diffractions from the individual crystals relative to the supporting carbon film, whose scattering is non-negligible for lower energy SEM electrons) and agglomeration (higher absorption of lower-energy SEM electrons within the agglomerates). Nevertheless, the recalculation of optimized TbF_3_ dataset with the new version of STEMDIFF package exhibited even higher improvement of the final powder diffraction pattern than in the case of strongly diffracting Au nanoislands—the diffractions much stronger, and the final diffractogram was much closer to the results from TEM/SAED (compare Figure 3b–d).

In analogy with the Au nanoislands, the positive effect of the improved 4D-STEM/PNBD method on the TbF_3_ diffraction pattern is best seen from the comparison of the 1D radially averaged diffraction profiles (Figure 3e). The calculation without PSF deconvolution (orange line) was sufficient to identify the TbF_3_ crystalline structure as the peak positions and intensities corresponded quite well to the TEM/SAED profile (black dotted line), but the resolution was rather low (just three broad maxima with weak shoulders at *q* ≈ 1.8, 3.3 and 5.2 Å^−1^). The calculation with PSF deconvolution (Figure 3e, red line) yield much better results: the first original broad peak (orange line, *q* ≈ 1.8 Å^−1^) was split into two sharp diffractions, the second peak (orange line, *q* ≈ 3.3 Å^−1^) was split to three diffractions, and the third low-intensity peak was split in three distinct maxima as well. Moreover, the PSF-deconvoluted profile showed an additional pair of very low intensity peaks at the highest diffraction angles (red line, *q* ≈ 6.5 Å^−1^). The resolution of the improved 4D-STEM/PNBD calculation was even slightly better in comparison with TEM/SAED (cf. red and black-dotted lines in Figure 3e).

The difference between the calculated PXRD pattern (Figure 3e, blue line) and all electron diffraction patterns (black, orange and red lines) was associated with strong preferred orientation (PO) of TbF_3_ nanocrystals, as explained elsewhere [18]. Briefly, the PXRD diffraction pattern was calculated for ideal, randomly oriented, isometric TbF_3_ nanocrystals, while real TbF_3_ synthesis resulted in thin nanoplatelets, most of which laid on their small facets oriented in such a way that their shortest unit cell parameter, *c*, was parallel with the electron beam [18]. This corresponded to the PO of the nanocrystals with zone axis [*uvw*] = [001]. Consequently, the Weiss zone law (WZL: *hu* + *kv* + *lw* = 0, where *h*,*k*,*l* are diffraction indexes and *u*,*v*,*w* are the indexes of the zone axis [22]) took quite simple form ([*uvw*] = [001] ⇒ *hu* + *kv* + *lw* = *h*0 + *k*0 + *l*1 = 0 ⇒ *l* = 0), indicating that the strongest diffraction peaks should be of the type [*hk*0]. Indeed, the experimental TEM/SAED and 4D-STEM/PNBD diffraction patterns (representing the real TbF_3_ nanoplatelets with strong PO) showed stronger *hk*0 diffractions (such as 020, 210 and 230) and weaker *hkl* diffractions (such as 111, 121 and 232) in comparison with the theoretically calculated PXRD diffraction pattern (representing the ideal crystals with random orientation). The small shift of the diffraction at *q* ≈ 3.3 Å^−1^ to the right after PSF deconvolution (red line vs. orange line in Figure 3e) could be attributed to the strong PO of TbF_3_ nanocrystals as well: the lower background after the deconvolution (cf. Figure 3b,d) lead to sharper and more precisely defined peak, whose maximum was shifted towards diffraction 230, in agreement with the WZL, as described above. The theoretical possibility that the shift might be a deconvolution artifact, is discussed below in Section 4.3.

#### 3.1.3. TiO_2_ Nanoparticles: Differentiation between Anatase and Rutile Modifications

The last two systems (TiO_2_ nanoparticles with two different crystalline modifications, anatase and rutile) were selected in order to demonstrate that the 4D-STEM/PNBD method can reliably distinguish nanocrystals with the identical chemical composition, but different crystalline structure. For the sake of brevity, Figure 4 summarizes just the final, radially averaged 1D diffraction profiles. For both anatase (Figure 4a) and rutile (Figure 4b), the agreement between theoretically calculated PXRD diffractograms (Figure 4, blue lines), experimental TEM/SAED diffractograms (Figure 4, black dotted lines) and experimental 4D-STEM/PNBD diffractograms calculated with PSF deconvolution (Figure 4, red lines) was very good. The intensity and resolution of diffractions in TEM/SAED and 4D-STEM/PNBD were almost identical. This confirmed the reliability and reproducibility of our improved 4D-STEM/PNBD method. The main conclusion was that the improved version of 4D-STEM/PNBD method could be employed in fast and easy identification of different nanocrystals with similar or even identical composition, which is impossible with other methods available in an SEM microscope.

### 3.2. Description of the Improved STEMDIFF Package

The freeware STEMDIFF program package converts the 4D-STEM dataset into 2D- and 1D-powder diffraction patterns, requiring just minimal user input. The substantially improved version of STEMDIFF, which was developed within this work, includes automatic entropy-based filtering, PSF deconvolution and many other improvements, such as the radial averaging of the final 2D diffraction pattern to the 1D diffraction profile. The final 1D diffractogram can be compared with the powder diffractograms from various databases, calculations and/or X-ray diffraction experiments. In the following subsections, we describe the basic STEMDIFF algorithm (Section 3.2.1) and the STEMIDFF user interface (Section 3.2.2), while the appendixes give information about the installation and efficient usage of the Python version of the package (Appendix A) and basic information about MATLAB version of the package (Appendix B).

#### 3.2.1. STEMDIFF Algorithm

The principal part of STEMDIFF algorithm is outlined in Figure 5. The scheme shows real data and outputs from the calculation of TbF_3_ nanocrystals (Section 3.1.2). The program always starts with a reading of a 4D-STEM dataset, i.e., with the reading of a 2D array of 2D diffractograms (2D nanobeam diffraction patterns, NBD) that are saved in the form of binary files (Figure 5a). The first processing route consists of a straightforward summation of all files (Figure 5b), but for most samples this is insufficient and the final radially averaged 1D-profiles (Figure 5c) contain broad, low-intensity peaks that are very hard to separate from the background.

The better processing of 4D-STEM/PNBD data employs entropy filtering: the program calculates Shannon entropy [23,24] of all files (Figure 5d), then it sums only the high-entropy files (Figure 5e), and after radial averaging we get well-defined peaks (Figure 5f). The principle of entropy-based filtering is illustrated in Figure 6: the lowest-entropy files contain just the central spot corresponding to the primary beam (Figure 6, upper left corner), with the increasing value of Shannon entropy (*S*) the files contain more-and-more diffraction spots (Figure 6, files with intermediate *S* values), and the highest-entropy files contain a high number of strong diffractions (Figure 6, lower right corner). The entropy-based filtering was the key part of the previous version of STEMDIFF package, which yielded 4D-STEM/PNBD diffractograms that were (after a careful background subtraction) comparable to TEM/SAED, although their resolution was somewhat lower [15].

The best processing of 4D-STEM/PNBD data, which has been included in the improved version of STEMDIFF package within this work, is based on PSF deconvolution: the program employs the lowest-entropy files in order to calculate 2D-PSF, which represents the XY-spread of the primary beam (Figure 5g). Then the program calculates PSF deconvolution of the individual high-entropy files before their summation (Figure 5h). After this treatment, the final radially averaged profiles contain sharp, high-intensity diffraction peaks (Figure 5i), whose resolution is fully comparable with the peaks from TEM/SAED, as exemplified in Section 3.

#### 3.2.2. STEMDIFF User Interface

The user interface of the recent version of STEMDIFF package is shown in Figure 7. The improved STEMDIFF package was developed as a part of this work. As illustrated in Figure 7, the current version of the package uses freeware Spyder integrated development environment (IDE) as a simple user interface (UI).

STEMDIFF package is focused on fast, easy and routine everyday use. The user just copies a template Python script (shipped within the program package) to Spyder (left pane in Figure 7) and modifies two key input parameters at the beginning of the script: the name of the directory with the input files and the number of iterations during PSF deconvolution. Optionally, it is possible to fine-tune some parameters that control the centering (i.e., finding the center of individual diffractograms) and summation, as described in Appendix A. When the script is run (Figure 7, green triangle in menu bar), the textual outputs are shown in Console window (Figure 7, lower right pane), and the final 1D-diffraction profile is shown in the Plots window (Figure 7, upper right pane). Other outputs (such as the 2D PNBD diffraction patterns shown in Figure 2, Figure 3 and Figure 5) are saved automatically in the current directory (i.e., in the directory with the modified template script).

### 3.3. Influence of Experimental and Processing Parameters on 4D-STEM/PNBD Results

#### 3.3.1. Data Processing Parameters: Summation and Deconvolution

Surprisingly enough, the data processing parameters exhibit a stronger influence on the quality of the 4D-STEM/PNBD diffractograms than the experimental parameters. In other words, if the data collection parameters are reasonable, the decisive factor influencing the final intensity and resolution of the diffractions is the way of summation and deconvolution within STEMDIFF package. This is illustrated in Figure 8 and Figure 9, which show final 1D-profiles of Au and TbF_3_ samples. For both samples, the data collection parameters (scanning step, scanning matrix, exposure time, etc.) were carefully optimized according to the recommendations summarized in Section 2.4.3. However, in both cases, the intensities and resolution of the diffractions were dramatically influenced by the summation type (Figure 8) and number of the deconvolution iteration cycles (Figure 9).

For the strongly diffracting Au nanoislands (Figure 8a and Figure 9a), the influence of the processing parameters was slightly lower (i.e., for this almost ideal sample we could get reasonable results even without entropy-based filtering and/or PSF deconvolution). Nevertheless, even in this case, the enhancement of peaks after deconvolution (Figure 8a), and with the increasing number of deconvolution iterations (Figure 9a) was strong and evident. The optimal number of deconvolution cycles was around 200, as the difference between 100 and 200 cycles (Figure 9a) was quite small and further deconvolution cycles just prolonged the computation time without apparent benefit.

For TbF_3_ nanoparticles with somewhat weaker diffractions (Figure 8b and Figure 9b), the straightforward summation of all NBD patterns without the entropy filtering and PSF deconvolution (Figure 8b, dotted line) yielded very broad and low-intensity peaks, which were almost impossible to separate from the background. The summation of just 20% of the files with the highest entropy (Figure 8b, orange line) resulted in two clear peaks (around 50 and 100 pixels from the center) and one very broad peak (ranging from ca 120 to 200 pixels from the center); this result was far from ideal, but sufficient for TbF_3_ identification (as documented in Figure 3e). The summation of high-entropy files after application of PSF deconvolution (Figure 8b, red line) enhanced both the intensity and resolution of the peaks drastically. The optimal number of deconvolution cycles for TbF_3_ nanocrystals (ca 300) was higher than for Au nanoislands (ca 200), which seems to be a general trend—lower diffracting samples require more deconvolution cycles.

#### 3.3.2. Experimental Parameters: Primary Beam Intensity and Exposure Time

The intensity of the primary beam and the exposure time are important parameters influencing 4D-STEM/PNBD results as illustrated in Figure 10. The measurement can be set in two principal ways: (i) non-overflown intensity of the diffractions (Figure 10a–c) or (ii) non-overflown intensity of the primary beam (Figure 10d–f).

The measurement of the first type (i.e., measurement R1; Figure 10a–c) was used in our previous work [15]. The maximal measurable intensity of our pixelated detector (*I*_MAX_) was 11,810 counts per pixel. The exposure time was adjusted in order to maximize intensities of diffracted beams while keeping them below *I*_MAX_. Under these conditions, the primary beam on the detector overflew (Figure 10a) and the summation of entropy-filtered files gave strong diffractions (Figure 10b). Nevertheless, it was not possible to reconstruct the correct XY-shape of the primary beam due to its overflown intensity (Figure 10a: the intensity was cut and the non-Gaussian shape of the primary beam after passing through the supporting carbon film could not be interpolated reliably). Therefore, the final summation with the PSF deconvolution failed to yield sharp diffractions (Figure 10c).

The optimized measurement of the second type (i.e., measurement R2; Figure 10d–f) was introduced in this study. The exposure time was adjusted so that the intensity of the primary beam was below *I*_MAX_ (Figure 10d). Consequently, the diffracted beams were somewhat weaker (Figure 10e) in comparison with measurement R1 (Figure 10b). On the other hand, it was possible to get the correct PSF of the primary beam and, as a result, the final summation including both entropy filtering and deconvolution (Figure 10f) gave much better results than measurement R1 (Figure 10c), which was the main advantage of the improved 4D-STEM/PNBD method.

#### 3.3.3. Experimental Parameters: Dataset Size

The effect of the dataset size on the quality of final 4D-STEM/PNBD diffraction patterns was quite small. This is illustrated in Figure 11 that shows the 1D-diffraction profiles of Au nanoislands. The diffraction profiles come from the simple summation of all files in order to eliminate possible effect from entropy filtering or PSF deconvolution. Figure 11 evidences that the final 1D diffraction profiles were almost independent on the number of datafiles for both high (Figure 11a) and low exposure times (Figure 11b).

## 4. Discussion

### 4.1. Originality and Novelty of 4D-STEM/PNBD Method

According to the available literature, which was summarized in the Introduction, our 4D-STEM/PNBD method is different from the existing 4D-STEM methods in three main aspects. At first, a great majority of other 4D-STEM methods have been developed in the field of transmission electron microscopy (4D-STEM-in-TEM; [8] and references therein), while our method is focused on scanning electron microscopy (4D-STEM-in-SEM; [15] and references therein). At second, the so-far-described 4D-STEM-in-SEM methods have been based on older, bulky CCD and CMOS detectors (which required rather non-standard hardware adjustments of SEM microscopes [11,12,13,14]), but our method relies on a modern, small DED detector (which can be installed to any modern SEM microscope with a standard port for STEM [10]). At third, both 4D-STEM-in-TEM and 4D-STEM-in-SEM methods found in the literature have been focused on the analysis of all individual NBD patterns of the 4D-dataset (which requires special software and special crystallographic knowledge [8,9,25]), but our method combines the individual NBD patterns into one simple PNBD pattern (which can be processed with standard, well-established, easy-to-use programs such as *PowderCell* [15,16,26] or *VESTA* [27]). Moreover, we have developed simple software requiring just a minimal user input (STEMDIFF package; Section 3.2), which automatically performs the summation of all individual NBD patterns in order to obtain the high-resolution PNBD pattern.

The focus of our 4D-STEM/PNBD method on powder diffraction analysis is its most distinctive feature. In TEM, some analogy of 4D-STEM/PNBD might have been developed and employed as well, but the standard, simpler and faster TEM/SAED method yields identical results directly (as exemplified in Section 3.1). Therefore, the 4D-STEM-in-TEM developers and users aim at analysis of the individual NBD patterns, although the possibility of combining selected NBD patterns into a powder diffractogram is occasionally mentioned and/or employed in calibration [9]. In SEM, some authors tested direct recording of PNBD patterns by means of special set of annular limiting apertures [11] or simple straightforward summation of individual NBD patterns during scanning [14], but our results showed unambiguously that this could work only for the nanocrystals with extremely strong diffractions (such as Au nanoislands in Figure 8a) and fails for systems containing smaller crystals (such as TbF_3_ in Figure 8b—note that the black dotted line representing the direct summation of NBD patterns does not show any distinct diffraction peaks). The difference between the traditional approach (straightforward summation of all files) and the recent version of 4D-STEM/PNBD method (optimized data collection combined with summation of high-entropy files with PSF deconvolution) is evident from the introductory Figure 1 as well: the traditional summation of all files from TbF_3_ dataset yielded just very broad, low-intensity diffraction peaks (Figure 1a), while the improvements introduced in the STEMDIFF package resulted in sharp, high-intensity diffractions (Figure 1d). Further analysis of the final, radially averaged 4D-STEM/PNBD diffraction patterns proved that their resolution is fully comparable or even slightly better in comparison with the common TEM/SAED diffractograms (Section 3.1.1, Section 3.1.2 and Section 3.1.3).

### 4.2. Advantages and Disadvantages of 4D-STEM/PNBD Method

As already emphasized in the introduction (Section 1) and description of STEMDIFF package (Section 3.2), the whole 4D-STEM/PNBD method is focused on a simple and routine everyday use. Nevertheless, the simplicity comes at certain cost. The reduction of the whole 4D-STEM dataset into a single 1D-powder diffractogram greatly simplifies data interpretation, but the information about the orientation of the individual nanocrystals within the sample is lost.

The main **advantages** of 4D-STEM/PNBD method are connected with its simplicity. We do not force potential users to analyze every single 2D-NBD pattern within the 2D-scanning array in order to get basic crystallographic information about the investigated sample. Instead, it is enough to perform substantially simpler analysis of one, averaged PNBD profile. Moreover, the final diffraction profile is obtained by means of a small, easy-to-understand Python script within freeware, easy-to-install STEMDIFF program package. This brings the following benefits:*Ease of use*: The measurement of 4D-STEM dataset in a modern SEM microscope with modern software is a routine task. The conversion of 4D-STEM data to 1D-powder diffractogram is performed automatically with freeware STEMDIFF package. The final analysis of a single 1D-powder electron diffractogram is usually quite easy as explained in the following item.*Accessible to everyone*: The method can be used by virtually anyone with basic SEM experience and elementary computer skills. The analysis of powder electron diffraction pattern is basically a fingerprint method consisting of three steps: (i) the obtaining experimental diffractogram by means of 4D-STEM/PNBD method as described above in Section 3.2, (ii) the simple calculation of the theoretical PXRD diffractogram of the analyzed sample by means of arbitrary free software, such as *PowderCell* [19] or *VESTA* [27], and (iii) the comparison of the results. If the experimental and theoretical diffractograms correspond to each other (such as those in Figure 2, Figure 3 and Figure 4), the crystal structure is identified.*The conversion of an SEM microscope to a powder diffractometer*: This opens quite new possibilities for SEM users. Standard SEM methods include imaging (with secondary, backscattered or transmitted electrons) and elemental analysis (such as energy-dispersive analysis of X-rays). A simple diffraction technique in an SEM microscope, equivalent to TEM/SAED, has been missing so far. The 4D-STEM/PNBD method aims to fill this gap as it enables to identify crystal structures of nanocrystalline powders like in the field of TEM microscopy.

The disadvantage of 4D-STEM/PNBD method results from the fact that it does not consider the additional information contained in the individual NBD patterns. More precisely, we use the individual NBD patterns just for the summation that yields the final PNBD pattern, while ignoring the specific positions and intensities of diffraction spots at each sample location. As the diffraction spots carry the information about the structure and orientation of the investigated material at a given location, the 4D-STEM/PNBD method cannot solve some special problems that are typically addressed by 4D-STEM-in-TEM methods, such as the visualization of different materials in the nanoscale (virtual imaging [28]), the identification of individual phases (structural classification [29]), the analysis of orientation of the individual nanocrystals (orientation mapping [30]), the analysis of strains of the individual nanocrystals (strain mapping [31]) and the enhancement of resolution and/or contrast of the micrographs by means of some advanced techniques (differential phase contrast or ptychography [32]). This is the price paid for the 4D-STEM/PNBD straightforwardness and simplicity.

### 4.3. Possible Artifacts on 4D-STEM/PNBD Diffractograms Connected with PSF Deconvolution

We have demonstrated that the PSF deconvolution dramatically enhances resolution of 4D-STEM/PNBD patterns, but a question may arise if the deconvolution cannot introduce some artifacts on the final powder diffractograms. The deconvolution artifacts cannot be excluded *a priori* as the deconvolution modifies the original diffractograms, but both the experimental results (showed above in Section 3.1) and theoretical considerations (discussed below in this section) suggest that in the case of 4D-STEM/PNBD method the deconvolution artifacts were minimized to a negligible level.

As for the experimental results (Section 3.1) the strongest argument documenting that the deconvolution artifacts are insignificant after our STEMDIFF processing, is as follows: the 4D-STEM/PNBD method was applied to four independent samples (Au nanoislands, TbF_3_ nanoplatelets, and two crystalline modifications of TiO_2_). The four resulting diffraction profiles (Figure 2e, Figure 3e, and Figure 4a,b) contained more than 40 diffraction peaks (ca 10 observable diffractions per pattern). For *all four diffraction patterns,* the positions and intensities of the peaks in the TEM/SAED profiles (non-deconvoluted diffractograms, black lines in Figure 2, Figure 3 and Figure 4) and 4D-STEM/PNBD profiles (deconvoluted diffractograms, red lines in Figure 2, Figure 3 and Figure 4) were in a very good agreement. The only small exception was observed in TbF_3_ diffraction profile (Figure 3e) showing a slight shift of the third diffraction peak at *q* ≈ 3.3 Å^−1^ to the right with respect to corresponding TEM/SAED peak (red vs. black line in Figure 3e). However, this shift could be attributed to the extremely strong preferred orientation (PO) of TbF_3_ nanoplatelets, which had already been described in Section 3.2.1. In this particular case, the PO was so strong that the (theoretically) strongest diffraction (111), which was not of the preferred *hk0* type, showed a minimum in both TEM/SAED and 4D-STEM/PNBD patterns. From this point of view, the low intensity of another non-preferred diffractions (002, 221, and 131) and the resulting shift of the third peak at *q* ≈ 3.3 Å^−1^ towards the preferred diffraction (230) confirmed that the deconvolution further improved the results in agreement with the observed PO effect. Yet another confirmation of the extremely strong PO in TbF_3_ sample was the negligible intensity of diffraction (232) at *q* ≈ 4.4 Å^−1^, which should be relatively strong according to theoretical PXRD calculation (blue line in Figure 3e), but it exhibited a very weak intensity in both TEM/SAED and 4D-STEM/PNBD patterns (black, orange and red lines in Figure 3e).

Available literature and theoretical considerations support the above-discussed experimental results in the sense that deconvolution artifacts are negligible in case of 4D-STEM/PNBD method. Firstly, image deconvolution is widely used in the field of light microscopy [33,34,35] and the artifacts are not considered critical. Secondly, our experimentally determined PSF is based on NBD patterns with the lowest entropy, which are dominated by low spatial frequencies and contain only the broad central spot of the primary beam as documented in Figure 6. The absence of sharp edges in both PSF (a convolution mask) and individual NBD patterns (source images for convolution, whose diffraction peaks are quite broad Gaussian-like functions as evidenced in Figure 2, Figure 3 and Figure 4, rather than sharp step functions) means that the possible *ringing artifacts* during the deconvolution are insignificant [36,37]. Thirdly, our algorithm eliminates possible artifacts due to small convolution mask, as the default STEMDIFF calculation employs a PSF function that is a few pixels larger in comparison with the original images (see parameters *psfsize* and *imgsize* in Figure 7; the detailed STEMDIFF documentation including an explanation of all parameters is available at www, as explained in Appendix A). The STEMDIFF calculation simply pushes the PSF border ring out of the area of interest.

### 4.4. Further Applications of 4D-STEM/PNBD Method

The specimens employed in this study were inorganic nanocrystals deposited on a thin carbon film. Analysis of nanocrystals is definitely useful by itself, but we plan to employ 4D-STEM/PNBD method also in characterization of other important types of samples: polymer nanocomposites (such as magnetic polymer microspheres filled with iron oxide nanoparticles [38]), inorganic nanoparticles in amorphous matrices (such as platinum nanoparticles in silica glass [39]), and biological samples containing nanocrystalline materials (such as calcite-precipitating bacteria [40] or biological tissues with nanocrystalline markers [41]). In these systems, a standard analysis of the inorganic nanoparticles by TEM/SAED is very difficult, as the majority of scattered electrons come from the amorphous matrix and diffractions from the nanocrystals are almost lost in the high background. The application of 4D-STEM/PNBD method should facilitate the analysis, because we could visualize the structure (using standard STEM/BF mode), select several locations with the nanocrystals, and collect signal only from the selected locations, ignoring the surrounding matrix. Preliminary results suggest that ca 10–20 nanocrystals with random orientations should be sufficient for the reliable identification of the crystalline structure by means of 4D-STEM/PNBD method.

## 5. Conclusions

The new version of our 4D-STEM/PNBD method converts arbitrary SEM microscope equipped with a pixelated STEM detector to a fast, high-resolution powder electron diffractometer. The 4D-STEM/PNBD method was designed for a simple, routine, and efficient everyday use with a minimal user input. It converts a huge 4D-STEM dataset, which is *easy to collect*, to a single powder diffraction pattern, which is *easy to process* even without deep crystallographic knowledge. This brings the classical *high-resolution* powder electron diffraction from the realm of TEM microscopy to the field of modern SEM microscopy and opens quite new possibilities for SEM users.

## Figures and Tables

**Figure 1 materials-14-07550-f001:**
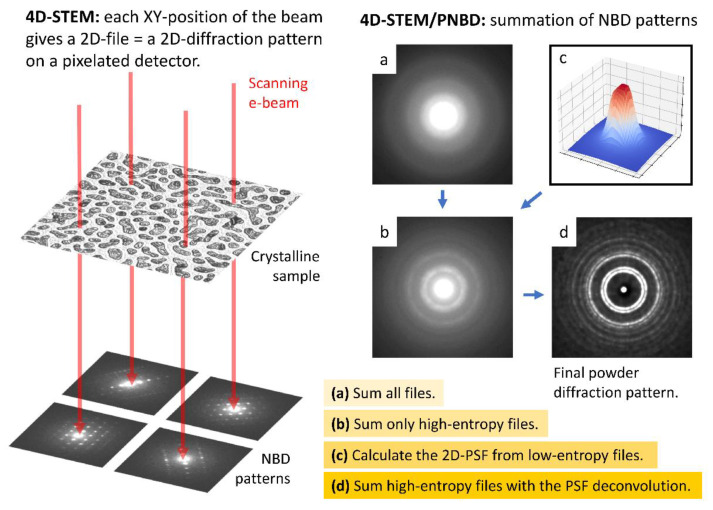
Principle of 4D-STEM/PNBD method. Left part of the image shows how the scanning electron beam penetrates through the sample and forms a set of nanobeam diffraction (NBD) patterns. Right part of the image shows how the individual NBD patterns can be combined into one powder nanobeam diffraction patterns (PNBD; with diffraction rings typical of the diffraction on polycrystalline samples): (**a**) PNBD obtained by the straightforward summation of all NBD patterns, (**b**) PNBD obtained by the summation of selected NBD patterns with high Shannon entropy values (high entropy indicated strongly diffracting crystals), (**c**) point spread function (PSF) calculated from low-entropy NBD patterns (the PSF represents the XY-spread of the primary beam), and (**d**) PNBD obtained by the summation of high-entropy NBD patterns with PSF deconvolution.

**Figure 2 materials-14-07550-f002:**
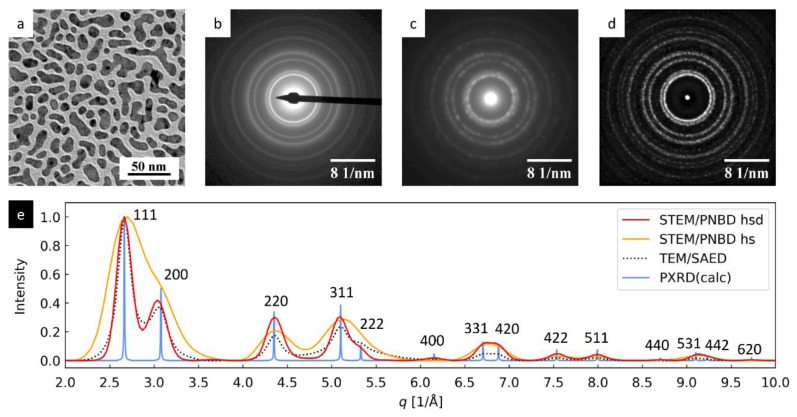
4D-STEM/PNBD results for Au nanoislands and their comparison with TEM and PXRD results: (**a**) TEM/BF image, (**b**) TEM/SAED diffraction pattern, (**c**) 4D-STEM/PNBD diffraction pattern calculated without PSF deconvolution, (**d**) 4D-STEM/PNBD pattern calculated with PSF deconvolution, and (**e**) comparison of radially averaged profiles from 4D-STEM/PNBD patterns with deconvolution (red line), 4D-STEM/PNBD without deconvolution (orange line), TEM/SAED (black dotted line), and theoretically calculated PXRD diffraction pattern of Au (blue line). All 4D-STEM/PNBD diffractograms and profiles in this figure were calculated for high-entropy files (20% of datafiles with the highest values of Shannon entropy as described in Section 2.4.2). The abbreviation *hs* and *hsd* in the plot legend mean calculation with high-entropy files (hs) and calculation with high-entropy files and PSF deconvolution (hsd), respectively.

**Figure 3 materials-14-07550-f003:**
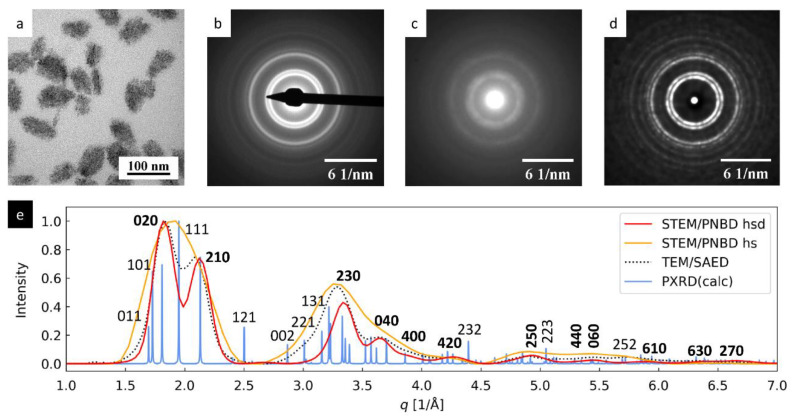
4D-STEM/PNBD results for TbF_3_ nanocrystals and their comparison with TEM and PXRD results: (**a**) TEM/BF image, (**b**) TEM/SAED diffraction pattern, (**c**) 4D-STEM/PNBD diffraction pattern calculated without PSF deconvolution, (**d**) 4D-STEM/PNBD pattern calculated with PSF deconvolution, and (**e**) comparison of radially averaged profiles from 4D-STEM/PNBD patterns with deconvolution (red line), 4D-STEM/PNBD without deconvolution (orange line), TEM/SAED (black dotted line), and theoretically calculated PXRD diffraction pattern of TbF_3_ (blue line). All 4D-STEM/PNBD diffractograms and profiles in this figure were calculated for high-entropy files (20% of datafiles with the highest values of Shannon entropy as described in Section 2.4.2). The abbreviation *hs* and *hsd* in the plot legend mean calculation with high-entropy files (hs) and calculation with high-entropy files and PSF deconvolution (hsd), respectively.

**Figure 4 materials-14-07550-f004:**
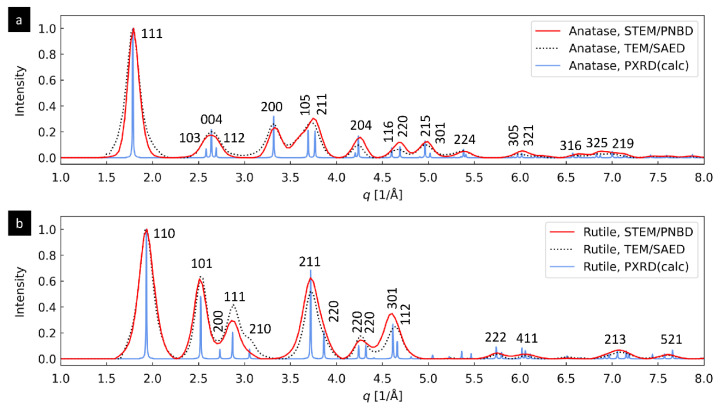
Comparison of 4D-STEM/PNBD, TEM/SAED and PXRD results for TiO_2_ nanocrystals with two different crystalline modifications: (**a**) anatase and (**b**) rutile. The 4D-STEM/PNBD results (red lines) were calculated with the recent version of the method including PSF deconvolution. The TEM/SAED results (black dotted lines) and PXRD results (blue lines) were obtained as described above (Section 2.1).

**Figure 5 materials-14-07550-f005:**
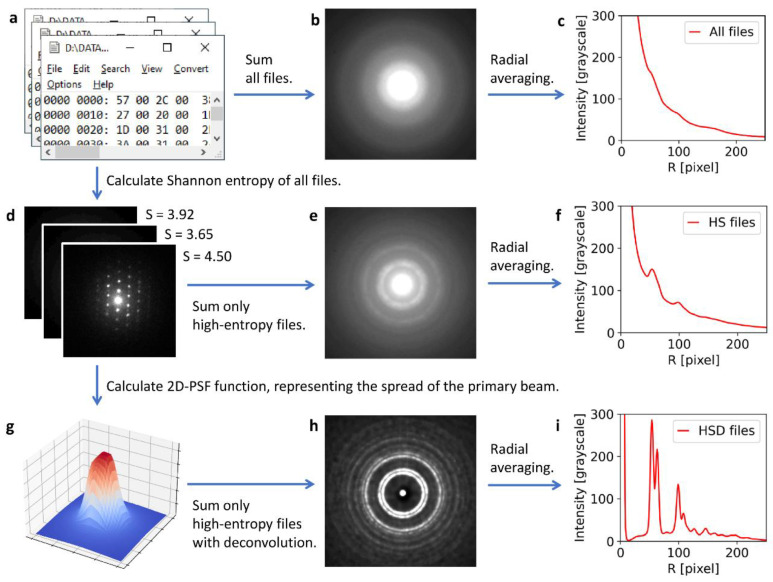
Scheme of STEMDIFF algorithm. The first processing route (**a**–**c**) is the straightforward summation of all files that contain the individual nanobeam diffraction patterns (**b**), but the final diffraction profiles contain very broad, low-intensity diffraction peaks (**c**). The second processing route (**d**–**f**) employs entropy-based filtering (**d**) and summation of just high-entropy files containing strong diffractions (**e**), which results in well-defined diffraction peaks, although their intensity and resolution is still suboptimal (**f**). The third processing route (**g**–**i**) includes determination of PSF, which represents the XY-spread of the primary beam (**g**), and application of PSF deconvolution to high-entropy files before their summation (**h**), which results in substantially sharper peaks with higher resolution and lower background (**i**).

**Figure 6 materials-14-07550-f006:**
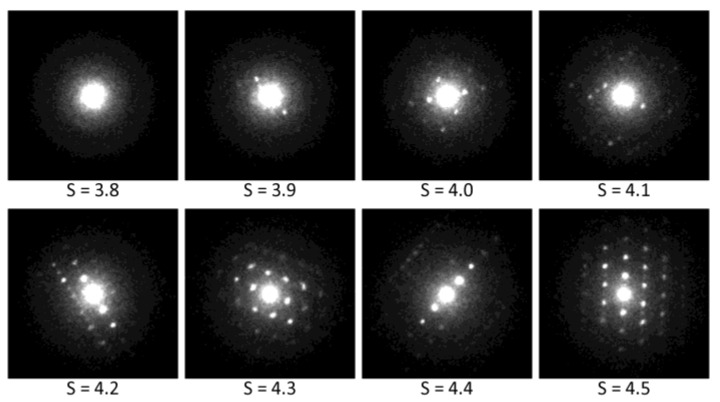
Entropy-based filtering: the datafiles with high Shannon entropy (*S*) contain strong diffraction peaks and *vice versa*.

**Figure 7 materials-14-07550-f007:**
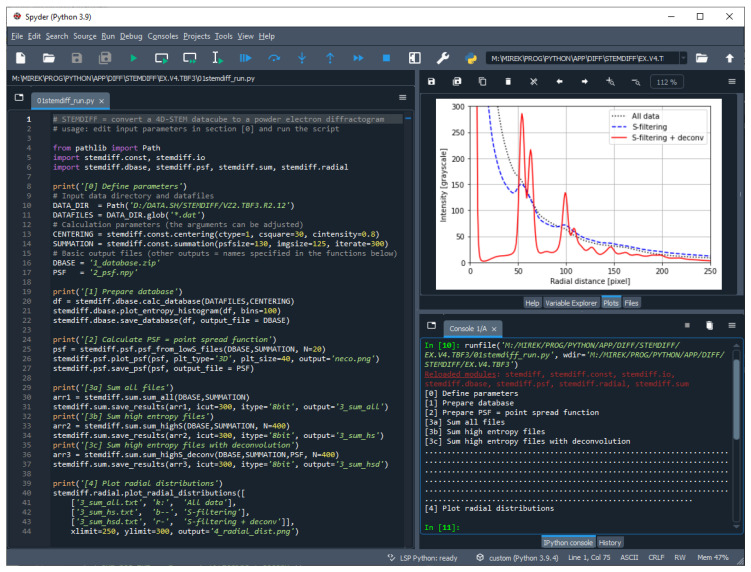
User interface of STEMDIFF package.

**Figure 8 materials-14-07550-f008:**
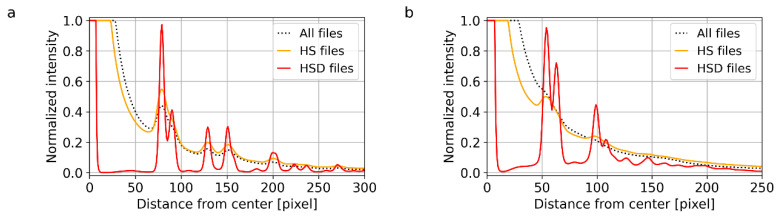
Influence of the summation type on the quality of 4D-STEM/PNBD results for (**a**) Au nanoislands and (**b**) TbF_3_ nanocrystals. Both plots show final 1D-diffraction profiles from STEMDIFF calculation. STEMDIFF package can sum all files (black dotted line), high-entropy files (HS files; orange line), or high-entropy files with PSF deconvolution (HSD files, red line).

**Figure 9 materials-14-07550-f009:**
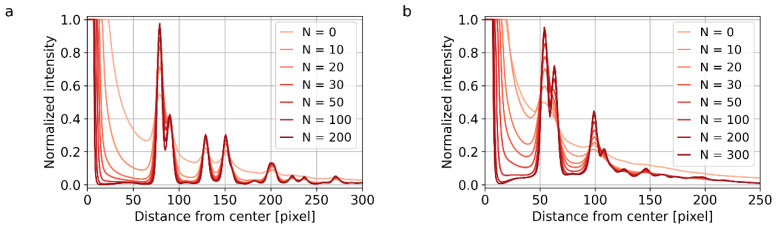
Influence of the number of iteration cycles, *N*, during PSF deconvolution on the quality of 4D-STEM/PNBD results for (**a**) Au nanoislands from Section 3.2.1 and (**b**) TbF_3_ nanocrystals from Section 3.2.2. Both plots show final 1D-diffraction profiles from STEMDIFF calculation.

**Figure 10 materials-14-07550-f010:**
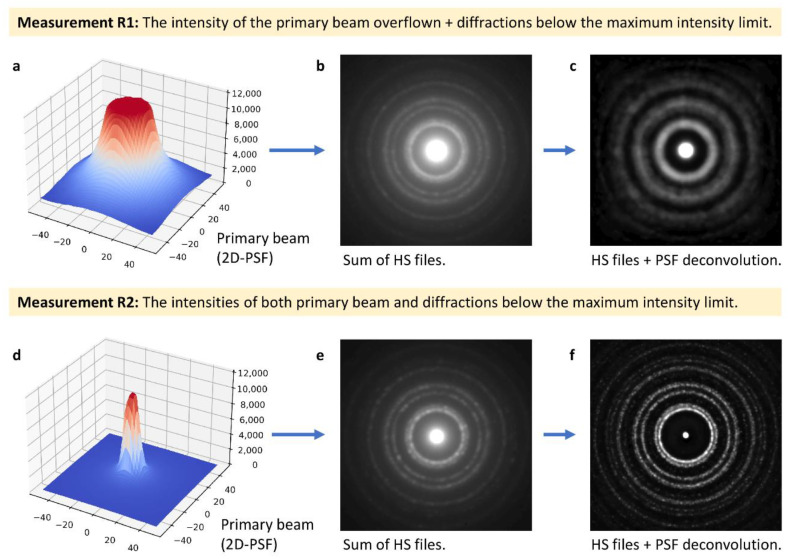
Influence of the measurement type on the quality of 4D-STEM/PNBD results for Au nanoislands. Measurement R1 (upper row) employs high exposure times, which results in (**a**) an overflown primary beam and (**b**) higher intensity diffractions, but (**c**) wrong PSF deconvolution results. Measurement R2 (lower row) employs lower exposure time, which leads to (**a**) a non-overflown intensity of the primary beam and (**b**) lower intensity of diffractions, but (**c**) correct PSF deconvolution results. All data shown in the figure come from the measurements of Au nanoislands. Measurement R1 (**a**–**c**) was made in our previous study [15], while the optimized measurement R2 (**d**–**f**) was carried out within this work.

**Figure 11 materials-14-07550-f011:**
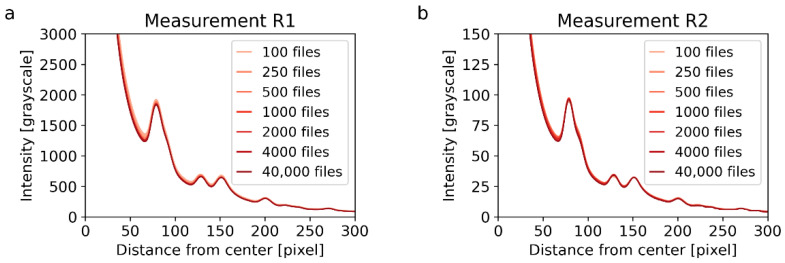
Influence of the dataset size on the quality of 4D-STEM/PNBD results. Both plots show 1D-diffraction profiles of Au nanoislands after summation of all files (i.e., entropy filtering and PSF deconvolution were not included in the calculation in order not to influence the results). The plots show the results of measurements R1 (**a**) and R2 (**b**). The measurements R1 and R2 differ by the exposure time as explained in Section 3.3.2 and illustrated in Figure 10.

**Table 1 materials-14-07550-t001:** Data acquisition settings.

Dataset ID	Step ^1^ [nm]	Scanning Matrix ^2^	HFW ^3^ [µm]	WD ^4^ [mm]	Probe Current [pA]	Total No. of Files	Duration [h:m:s]
Au	20	9 × 230	4.6	5.4	25	2070	0:05:30
TbF_3_	50	12 × 190	9.5	4.8	25	2280	0:06:20
TiO_2_/anatase	25	20 × 100	2.5	3.2	13	2000	0:10:30
TiO_2_/rutile	25	20 × 100	2.5	3.2	13	2000	0:10:30

^1^ Step = scanning step = distance between two scanning points. ^2^ Scanning matrix = rectangular XY array of measurement points. ^3^ HFW = Horizontal field width = real width of the STEM/BF image. ^4^ WD = Working distance = distance between electron column pole-piece and sample.

## Data Availability

All data (STEM and TEM micrographs, 4D-STEM data cubes) are available at request to the first author (M.S.).

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
