# Peer review of "High Resolution Powder Electron Diffraction in Scanning Electron Microscopy"

_materials, 2021, doi:10.3390/ma14247550_

Round 1

Reviewer 1 Report

The authors took into account my remarks to the previous version of the manuscript as much as possible and I have no further remarks.

Author Response

Reviewer 1: The authors took into account my remarks to the previous version of the manuscript as much as possible and I have no further remarks.

Answer: We thank the reviewer for the positive evaluation and also for his/her previous comments that helped us improve the manuscript.

Reviewer 2 Report

The manuscript ID materials-1493244 mainly presents particular measurements and a development associated with a four-dimensional scanning transmission electron microscopy technique assisted by a powder nanobeam diffraction. Here are some points to the authors:

  1. From the text should be clear how this report represents a full study instead of a fragmentation of work integrated by multiple manuscripts.
  2. The findings should be presented not as incremental but as an advance to the field underlying materials science principles.
  3. A few citations of updated publications are presented. Especially for the discussion section.
  4. It would be interesting if the authors could extend some comments for the assistance of this research in other microscopy techniques. You can consider for instance studies involving biological system (https://doi.org/10.3390/nano11051227)or magnetism occurring only in nanoscale samples (doi:10.1017/S1431927619015204).
  5. Evidences of the standard characterization of the samples studied are missing in the report.
  6. If possible, please report about the statistical analysis of the samples.
  7. Please report about the reproducibility in the measurements and error bar in the evaluations.
  8. It is not clear the systematic selection of some of the parameters employed in this research.
  9. I consider that the word “microscope” in the title could be changed for “microscopy” taking into account that this is not a technical design.
  10. The collective citations should be split in order to see the justification of each selected reference to be mentioned in this work.

Author Response

Point-by-point answers to the 2nd reviewer comments are in the attached file.

Round 2

Reviewer 2 Report

The authors have succesfully clarified all the points raised in the review stage. The results are relevant and the analysis is solid. In my opinion, this work worth publication in present form.

This manuscript is a resubmission of an earlier submission. The following is a list of the peer review reports and author responses from that submission.

Round 1

Reviewer 1 Report

Electron diffraction in the 4D-STEM mode in an SEM is certainly a topic of interest as many more labs have a SEM than an TEM.  However, as I understand from the paper, after turning the data into a powdr profile, the only information obtained from the data is the same as from typical X-ray powder diffraction data.  One could argue that in SEM you can choose to take only certain particles, as explained in the paper: you can image the particles and then take diffraction from only the zones of interest and ignore the surroundings. However, how then can you be sure that you get statistically relevant amounts of data if you have to start manually selecting the crystals (there was no example of such case in the paper).

It seems to me all examples were made on single phase samples and all obtained diffraction patterns from all area were added.  In this case, what is the advantage over powder XRD? How would you go about if you have a mixture of phases? If you cannot determine in any other way on the SEM the difference (as stated by the authors themselves) between for example anatase and rutile, then you cannot divide the crystals into separate groups before adding and they would just overlap in the powder diffraction pattern.

Also, currently the figures only show a comparison with theoretically calculated XRD, which have nice sharp peaks, but what would an experimental pattern of such small particles look like?   Would the PSF filtered SEM data presented here have sharper peaks than lab-powder XRD?

The remarks on installation and tips and tricks are not really necessary in the paper, this is something that belongs on the webpage where one can download the software, or at most in the Supporting information, but not in the main text. The text is rather long as it is.

Reviewer 2 Report

Slouf et al demonstrated using pixelated detector in SEM to record powder diffraction pattern by summing all 4D-STEM data. They had demonstrated the same setup in their previous publication (ref 15). The only new message in the current manuscript is they improved the sharpness of the diffraction pattern by deconvolving the point spread function (PSF) from the individual 4D-STEM patterns.

I have two major disagreements for publishing this article in nanomaterials.

  1. The major improvement that the authors brought is the deconvolution of the diffraction patterns. It did improve the sharpness of the data but I consider it is a small increment. This can easily be described in a short letter rather than a long article with >20 pages, where many statements are very similar repeats of the author’s previous paper (ref 15). The development of adding the deconvolution is quite trivial and had been applied in TEM diffraction (e.g. McBride et al, Ultramicroscopy, 94 (2003) p305-308). Without any new discovery in the materials, such a small point is far from the merit for publishing to nanomaterials.
  2. The position of diffraction rings in the deconvolution results shows unacceptable deviation from the TEM diffraction and calculated XRD. Such as, the 2nd peak around 3 1/Å in Fig 2e and the 2nd peak around 2.2 1/Å and particularly the 3rd peak around 3.3 1/Å in Fig 3e are explicitly deviated from both of the TEM diffraction and PXRD. Especially the 3rd peak of Fig 3e is largely right shifted comparing to the SEM/PNBD-hs which are without the convolution. The authors simply neglected these inaccuracies in the manuscript. The peak position is the most primary information of the powder diffraction, as the d-spaces are expected to be obtained quantitatively from the powder diffraction experiment. What are the error originated from and how they correlate to the artifacts usually happen in the deconvolution? These are not described in the manuscript.

This work is more suitable for a microscopy journal. I also suggest the authors re-write the manuscript in the concise way by referring their previous paper (ref 15).